# Scanning of Bridge Surface Roughness from Two-Axle Vehicle Response by EKF-UI and Contact Residual: Theoretical Study

**DOI:** 10.3390/s22093410

**Published:** 2022-04-29

**Authors:** Y. B. Yang, Baoquan Wang, Zhilu Wang, Kang Shi, Hao Xu

**Affiliations:** 1National Engineering and Research Center for Mountainous Highways, Chongqing 400067, China; ybyang@cqu.edu.cn; 2School of Civil Engineering, Chongqing University, Chongqing 400044, China; bqwang@cqu.edu.cn (B.W.); shikang@cqu.edu.cn (K.S.); hxu@cqu.edu.cn (H.X.); 3School of Civil Engineering and Architecture, Chongqing University of Science and Technology, Chongqing 401331, China; 4Xi’an CCCC Civil Engineering Technology Co., Ltd., Xi’an 710075, China

**Keywords:** bridge, Kalman filter, residual response, road profile, vehicle, vehicle scanning method

## Abstract

The scanning of bridge surface roughness by the test vehicle is a coupled and non-stationary problem since the bridge deflection caused by vehicles will inevitably enter into the vehicle response. To this end, a two-step procedure is proposed to retrieve the bridge surface profile from the noise-contaminated responses of a two-axle vehicle moving over bridges. Central to this is the elimination of the bridge deflection from the estimated unknown input to the test vehicle system. First, the extended Kalman filter with unknown inputs (EKF-UI) algorithm is extended to formulating the state-space equations for the moving vehicle over the bridge. Analytical recursive solutions are derived for the improved vehicle states and the unknown input vector consisting of the vehicle–bridge contact displacement and surface profile. Second, the correlation between the cumulated contact residuals and contact displacements for the two axles is approximately defined by using the vehicle’s parameters and location on the bridge. Then, the surface profile is retrieved from the unknown input by removing the roughness-free contact (bridge) displacement, calculated with no prior knowledge of bridge properties. The efficacy of the proposed procedure was validated by the finite element method and demonstrated in the parametric study for various properties of the system. It is confirmed that the retrieved bridge surface profile is in excellent agreement with the original (assumed). For practical use, the vehicle is suggested to run at a not-too-high speed or in a too noisy environment. The proposed technique is robust with regard to vehicle mass and bridge damping.

## 1. Introduction

Based on vehicle–bridge interaction (VBI) dynamics [1,2,3], the moving test vehicle has been proposed to measure the dynamic properties of bridges, which is known as the vehicle scanning method (VSM) for bridges [4,5]. In this regard, pavement roughness is known to be a crucial factor that may affect the vibration characteristics of the VBI system in general and the spectral analysis of the vehicle response in particular [6,7,8]. To this end, many techniques have been presented for the purpose of alleviating the roughness effect on using the vehicle response for bridge property detection, such as wavelet transform [9,10,11], filtering techniques [12,13], machine learning approach [14], dual connected vehicles [15,16,17,18], external excitation [19,20,21,22,23], etc. It is believed that the ability to predict pavement roughness is essential to the understanding of the global VBI mechanism, while deepening VSM technology specifically. In addition, roughness has often been used by highway engineers to assess the health condition of bridges for maintenance arrangements. Hence, there is a practical need for efficient techniques for estimating pavement roughness.

Conventional techniques for measuring road surface roughness can be classified into contact and non-contact measurements [24,25,26,27]. The contact techniques generally utilize two types of devices: manual profilograph, e.g., rods and levels, walking profilers, and trailer-towed devices, e.g., the longitudinal profile analyzer, etc. Non-contact techniques usually adopt laser profilers, light detection and ranging (LiDAR) systems, and three-dimensional (3D) image devices. Although both techniques show some advantages in measurement accuracy, they are generally time-consuming in operation, inefficient for long-distance measurement, and expensive in equipment acquisition and maintenance. As such, they are not suitable for large-scale applications or, particularly, for the regular monitoring of bridges. Recently, eigen perturbation real-time techniques were presented for long-term monitoring of vibrating systems [28,29], which were reported to be effective in detecting minor road cracks and robust toward both vibration and image-based measurements.

Using the VSM to detect pavement roughness, only one or a few sensors need to be installed on the test vehicle for bridge scanning, which is advantageous for its relatively low cost and high efficiency and mobility. Imine and Delanne [30] proposed using the vertical response of a full vehicle model to identify road roughness based on the sliding mode observation algorithm. Ngwangwa et al. [31] recovered the road roughness using either a single-axle vehicle based on the artificial neural network (ANN) algorithm or a two-axle vehicle based on the Bayesian regularized nonlinear autoregressive exogenous model. In addition to the ANN algorithm, Yousefzadeh et al. [32] estimated road roughness by a full vehicle model using the software of Automated Dynamic Analysis of Mechanical Systems (ADAMS). González et al. [33] classified road roughness based on the transfer function between the road roughness and the vertical response of the instrumented vehicle. Harris et al. [34] characterized pavement profile heights using the acceleration response of a moving vehicle and a combinatorial optimization technique. Qin et al. [35] used the acceleration response of the unsprung mass of a single-axle vehicle to identify road surface roughness. Wang et al. [36] utilized the data collected by tire pressure sensors and the transfer function method to estimate road roughness.

In addition, the Kalman filter (KF) was also applied to estimate road roughness from the vehicle response. For general road pavements, Doumiati et al. [37] studied a real-time method of using the suspension deflection and body acceleration response of a single-axle vehicle to measure road surface roughness based on the KF procedure. Wang et al. [38] proposed the combined use of the Minimum Model Error criterion and the Kalman filter algorithm to improve the estimation of the road profile for a vehicle suspension system. In the work by Kang et al. [39], the discrete Kalman filter (DKF) with unknown input was used instead. Kim et al. [40] designed an improved discrete Kalman filter to simultaneously estimate unknown road roughness input and state variables for a vehicle suspension control system.

However, for the scanning of bridge surface roughness by the test vehicle, the bridge oscillation caused by the vehicles will inevitably enter into the vehicle response, which makes the problem coupled and non-stationary and different from the road surface detection (which is basically stationary). With respect to bridge surface roughness, Wang et al. [41] used the particle filter technique to estimate bridge roughness from the response of a moving vehicle considering the VBI effect. Zhan and Au [42] estimated bridge roughness by letting a test vehicle pass the bridge multiple times with different added masses, which allows elimination of bridge displacement. Based on a minimum variance unbiased estimator with an optimization scheme, Shereena and Rao [43] estimated bridge roughness with the VBI effect considered. Yang et al. [44] identified the bridge roughness by two-connected vehicles by using the displacement influence lines to eliminate the approximate correlation of the deflections of the two contact points. Even though the VBI has been considered in the above research, the bridge deflection elimination was reported to be incomplete for roughness calculation. Moreover, the noise in the measurement was not fully accommodated.

To tackle these problems, this study proposed a two-step technique to estimate bridge roughness using vehicle response in a noisy environment. In the first step, the extended Kalman filter with unknown inputs (EKF-UI) algorithm proposed by J.N. Yang in 2007 [45] was employed to generate the unknown input (external excitation) to the test vehicle based on the noise-contaminated vehicle response. In the second step, the surface roughness profile is retrieved from the unknown input by removing the bridge deflection based on the contact residual.

In the development of a new approach, numerical validation or simulation should be conducted before any field validation can be carried out. In this study, the numerical validation consists of two phases that are of completely different algorithms: (1) Forward phase: The random original (or input) surface profile generated by codes is used as input to the VBI calculation program to simulate the dynamic response of the test vehicle moving over a rough bridge. Next, Gaussian white noise will be superimposed on the vehicle responses to simulate the (measured) noise-contaminated vehicle response. (2) Backward phase: The proposed two-step technique will be applied to the noise-contaminated vehicle response created in the forward phase to estimate the improved (noise-reduced) vehicle states and surface profile.

This paper is organized as follows. In Section 2, a brief is given of the EKF-UI algorithm. Section 3 describes the two-step technique for retrieving bridge surface profiles. In Section 4, the surface profile generation and the finite element method (FEM) for simulating the VBI system are summarized, together with the efficacy of the proposed procedure validated. In Section 5, a parametric study is carried out for the proposed procedure against various factors. Finally, the conclusions are drawn in Section 6.

## 2. Extended Kalman Filter with Unknown Inputs (EKF-UI)

Kalman filters and their variants, such as extended KF (EKF), adaptive KF (AKF), unscented KF (UKF), etc., have been developed to meet various needs. It is known that they differ in the level of accuracy, but the primary difference is in the applicability to the type of problems faced [46]. This paper deals with the problem of using the measured response (non-stationary) of the moving test vehicle over a rough bridge (an oscillating elastic structure) to estimate the signal from the bridge to the test vehicle, of which the vehicle–bridge interaction is of major concern. This problem differs from the one encountered in road roughness detection. The input of the moving test vehicle consists of two parts, i.e., the random space-varying pavement roughness and the time-varying bridge deflection caused by the test vehicle; both are unmeasurable and unavailable, i.e., unknown to the test vehicle. Obviously, the KF-based algorithms, including the KF, EKF, UKF and AKF, are no longer inapplicable, as they require all the external input data (excitation data) to be measured or available. To this end, the EKF-UI algorithm proposed by Yang [45] was adopted, since it can be effectively used to estimate the unknown input to a system based on the measured output data. For example, seismic analysis allows us to identify the unknown ground motions from the measured structural responses.

As stated, this section forms the backward phase of the numerical study. The following is a brief description of the EKF-UI algorithm. For a linear system with unknown inputs, the equation of motion is
(1)$$M\ddot{x}\left(t\right)+C\dot{x}\left(t\right)+Kx\left(t\right)=\eta^{*}f^{*}\left(t\right)+\eta f\left(t\right)$$
where **M**, **C** and **K** denote the mass, damping and stiffness matrices of the system, respectively; $\ddot{x}\left(t\right)$,$\dot{\;x}\left(t\right)$, and x(t) the acceleration, velocity and displacement responses; $f^{*}\left(t\right)$ and f(t) the unknown and known input vectors, respectively; and $\eta^{*}$ and η the corresponding influence matrices for $f^{*}\left(t\right)$ and f(t). The vector $f^{*}\left(t\right)$ is the input force or quantity of motion (displacement, velocity and acceleration) [45], which indicates the rise and fall of bridge pavement input to the vehicle in this study, as will be demonstrated in Section 3.1. By defining the state vector as $Z\left(t\right)={\left\{{\dot{x}}^{T},x^{T}\right\}}^{T}$, Equation (1) can be transformed into the discrete-time state space as:(2)$$Z_{k+1}=A_{k}Z_{k}+B_{k}f_{k}+B_{k}^{*}f_{k}^{*}+w_{k}$$
where $A_{k}$ is the state transition matrix, and$\;B_{k}$ and $B_{k}^{*}$ the influence matrices of the known and unknown inputs $f_{k}$ and $f_{k}^{*}$, respectively; $w_{k}$ is the noise vector caused by system uncertainty, which has a zero mean and a covariance matrix as $Q_{k}$, i.e., $E\left[w_{k}\right]=0\;$and $E\left[w_{k}\;\;w_{k}{}^{T}\;\right]=Q_{k}$.

The discrete measurement equation of the system can be expressed as:(3)$$y_{k+1}=C_{k+1}Z_{k+1}+D_{k+1}f_{k+1}+D_{k+1}^{*}f_{k+1}^{*}+v_{k+1}$$
where $y_{k+1}={y\left(t\right)|}_{t=\left(k+1\right)T_{s}}$ denotes the measurement vector, $C_{k+1}$ the measurement matrix, $D_{k+1}$ and $D_{k+1}^{*}$ the influence matrices of the known input $f_{k+1}$ and unknown input $f_{k+1}^{*}$, respectively, on the measurement vector $y_{k+1}$; and $v_{k+1}$ the measurement noise vector, assumed to be a Gaussian white noise with zero mean and a covariance matrix as $R_{k}$, i.e., $E\left[v_{k}\right]=0$ and $E\left[v_{k},v_{k}^{T}\right]=R_{k}$.

Let ${\hat{Z}}_{k+1|k+1}$ and ${\hat{f}}_{k+1|k+1}^{*}$ be the improved estimates of the state vector $Z_{k+1}$ and unknown input vector $f_{k+1}^{*}$, respectively. Based on the EKF-UI algorithm, they can be obtained recursively by following the flowchart given in Figure 1 and the steps presented in Appendix A.

## 3. Formulation of the Problem of Concern with EKF-UI

In this study, the moving test vehicle will be fitted with the proper vibration sensors. The unknown input for the test vehicle system consists of two parts: one is the surface roughness, and the other is the bridge deflections caused by vehicles, including the scanning test vehicle. The novelty of this study is the proposal of a two-step procedure to retrieve the “pure” bridge surface roughness profile from the noise-contaminated responses of a two-axle vehicle, while eliminating the bridge deflection. In this section, the formulation of the proposed technique will be presented. First, the procedure of using a moving two-axle test vehicle to generate the unknown input by the EKF-UI will be presented. Then, the contact displacements calculated from the roughness-free contact residuals will be derived and deducted from the unknown input to yield the bridge surface profile. The idea proposed herein can be equally applied to test vehicles with multi axles.

### 3.1. Vehicle–Bridge Interaction (VBI) Model for Retrieving Surface Profile

For the monitoring of bridges, the use of a test vehicle with two axles is more convenient than the one with a single axle for its ability to self-stand. Moreover, the residual response generated by the two axles is roughness free, which will be utilized in retrieving the surface profile. Consider a two-axle test vehicle moving over a simply supported bridge, as shown in Figure 2. The vehicle (body) is simulated as a rigid beam of mass $m_{v}$ and moment of inertia $J_{v}$, and supported by two springs spaced at d and of stiffnesses $k_{1}$ and $k_{2}$. The vehicle is *asymmetric* in that its center of gravity, C, is unequally spaced from the front axle A_1_ and rear axle A_2_, i.e., with distances $d_{1}$ and $d_{2}$, respectively. Consequently, two DOFs are needed for the vehicle to simulate its vertical and rotational motions $y_{v}$ and $\theta_{v}$. The bridge is modeled as a Bernoulli–Euler beam of span length *L*, elastic modulus *E*, moment of inertia *I*, and mass per unit length *m*. For the test vehicle, only the acceleration responses are of concern, since they can be easily measured in practice. The vehicle damping is ignored in the analytical formulation, for a good test vehicle is to be designed with the least damping for better transmissibility. However, bridge’s damping will be included in the finite element simulation as it cannot be ignored.

The test vehicle in movement will interact with the bridge via the two contact points P_1_ and P_2_ as in Figure 2, and in turn be set in vertical motion by the vibrations transmitted upward from the bridge. The vibration of each contact point is composed of two parts, i.e., the contact displacement u and the surface profile r of the bridge. For the vehicle with its front axle acting at x (=vt), the equations of motion can be written in terms of the vertical displacement $y_{v}\left(t\right)$ and rotational angle $\theta_{v}\left(t\right)$ as
(4)$$\begin{array}{c} m_{v}{\ddot{y}}_{v}\left(t\right)+k_{1}\left\{y_{v}\left(t\right)+d_{1}\theta_{v}\left(t\right)-\left[u_{1}\left(t\right)+{r\left(x\right)|}_{x=vt}\right]\right\}\\ \;\;\;\;\;\;\;\;\;\;\;\;\;\;\;\;\;+k_{2}\left\{y_{v}\left(t\right)-d_{2}\theta_{v}\left(t\right)-\left[u_{2}\left(t\right)+{r\left(x-d\right)|}_{x=vt}\right]\right\}=0 \end{array}\;\begin{array}{c} \;J_{v}{\ddot{\theta }}_{v}\left(t\right)+d_{1}k_{1}\left\{y_{v}\left(t\right)+d_{1}\theta_{v}\left(t\right)-\left[u_{1}\left(t\right)+{r\left(x\right)|}_{x=vt}\right]\right\}\\ \;\;\;\;\;\;\;\;\;\;\;\;\;\;\;\;-d_{2}k_{2}\left\{y_{v}\left(t\right)-d_{2}\theta_{v}\left(t\right)-\left[u_{2}\left(t\right)+{r\left(x-d\right)|}_{x=vt}\right]\right\}=0 \end{array}$$
where ${\ddot{y}}_{v}\left(t\right)\;$and ${\ddot{\theta }}_{v}\left(t\right)$ are accelerations; $u_{1}\left(t\right)$ and$\;u_{2}\left(t\right)\;$the displacements of the bridge at the contact points P_1_ and P_2_; and ${r\left(x\right)|}_{x=vt}$ and ${r\left(x-d\right)|}_{x=vt}$ the corresponding surface profile.

As shown in Figure 2, the vertical displacement $y_{v}\left(t\right)$ and rotational angle $\theta_{v}\left(t\right)$ of the vehicle can be related to the car body responses $y_{1}$. and $y_{2}$ at the two points A_1_ and A_2_ as follows:(5)$$y_{v}=\frac{d_{1}y_{2}+d_{2}y_{1}}{d}\;,\theta_{v}=\frac{y_{1}-y_{2}}{d}$$

Substituting Equation (5) into Equation (4) yields the equations of motion in terms of the car body displacements $y_{1}$ and $y_{2}$ as
(6)$$\;\begin{array}{cc} \frac{m_{v}}{d}\left[d_{1}{\ddot{y}}_{2}\left(t\right)+d_{2}{\ddot{y}}_{1}\left(t\right)\right]+k_{1}\left\{y_{1}\left(t\right)-\left[u_{1}\left(t\right)+{r\left(x\right)|}_{x=vt}\right]\right\} & \\ +k_{2}\left\{y_{2}\left(t\right)-\left[u_{2}\left(t\right)+{r\left(x-d\right)|}_{x=vt}\right]\right\} & \;=\;0 \end{array}\;\begin{array}{cc} \;\frac{J_{v}}{d}\left[{\ddot{y}}_{1}\left(t\right)-{\ddot{y}}_{2}\left(t\right)\right]+d_{1}k_{1}\left\{y_{1}\left(t\right)-\left[u_{1}\left(t\right)+{r\left(x\right)|}_{x=vt}\right]\right\} & \\ -d_{2}k_{2}\left\{y_{2}\left(t\right)-\left[u_{2}\left(t\right)+{r\left(x-d\right)|}_{x=vt}\right]\right\} & \;=\;0 \end{array}$$

By performing the following operations to Equation (6): “the *first* one” $\times J_{v}+$“the *second* one”$\times d_{1}m_{v}$ and “the *first* one” $\times J_{v}-$“the *second* one”$\times d_{2}m_{v}$, one can arrive at the following two equations:(7)$$\begin{array}{c} m_{v}J_{v}{\ddot{y}}_{1}\left(t\right)+k_{1}\left(J_{v}+d_{1}^{2}m_{v}\right)\left\{y_{1}\left(t\right)-\left[u_{1}\left(t\right)+{r\left(x\right)|}_{x=vt}\right]\right\}\\ \;+k_{2}\left(J_{v}-d_{1}d_{2}m_{v}\right)\left\{y_{2}\left(t\right)-\left[u_{2}\left(t\right)+{r\left(x-d\right)|}_{x=vt}\right]\right\}=0\\ m_{v}J_{v}{\ddot{y}}_{2}\left(t\right)+k_{1}\left(J_{v}-d_{1}d_{2}m_{v}\right)\left\{y_{1}\left(t\right)-\left[u_{1}\left(t\right)+{r\left(x\right)|}_{x=vt}\right]\right\}\\ \;+k_{2}\left(J_{v}+d_{2}^{2}m_{v}\right)\left\{y_{2}\left(t\right)-\left[u_{2}\left(t\right)+{r\left(x-d\right)|}_{x=vt}\right]\right\}=0 \end{array}$$
which can be expressed in matrix form as
(8)$$M\ddot{x}\left(t\right)+Kx\left(t\right)=\eta^{*}f^{*}\left(t\right)$$
where
(9)$$x={\left[y_{1}y_{2}\right]}^{T},M=\left[\begin{array}{cc} m_{v} & 0\\ 0 & m_{v} \end{array}\right]\;K=\left[\begin{array}{cc} k_{1}\frac{\left(J_{v}+d_{1}^{2}m_{v}\right)}{J_{v}} & k_{2}\frac{\left(J_{v}-d_{1}d_{2}m_{v}\right)}{J_{v}}\\ k_{1}\frac{\left(J_{v}-d_{1}d_{2}m_{v}\right)}{J_{v}} & k_{2}\frac{\left(J_{v}+d_{2}^{2}m_{v}\right)}{J_{v}} \end{array}\right]\;\eta^{*}=\left[\begin{array}{cc} k_{1}\frac{\left(J_{v}+d_{1}^{2}m_{v}\right)}{J_{v}} & k_{2}\frac{\left(J_{v}-d_{1}d_{2}m_{v}\right)}{J_{v}}\\ k_{1}\frac{\left(J_{v}-d_{1}d_{2}m_{v}\right)}{J_{v}} & k_{2}\frac{\left(J_{v}+d_{2}^{2}m_{v}\right)}{J_{v}} \end{array}\right]\;f^{*}\left(t\right)=\left[\begin{array}{c} u_{1}\left(t\right)+{r\left(x\right)|}_{x=vt}\\ u_{2}\left(t\right)+{r\left(x-d\right)|}_{x=vt} \end{array}\right]$$

Here$\;m_{v}$, $J_{v}$, $k_{1}$,$\;k_{2}$, $d_{1}$, $d_{2}$ and d are all the properties of the two-axle vehicle that are available prior to the test, ${\ddot{y}}_{1}$ and ${\ddot{y}}_{2}$ are the measured car body accelerations, and $y_{1}$ and $y_{2}$ can be integrated from the corresponding accelerations [47,48]. Note that in the double integration of vehicle’s acceleration for displacement, low-frequency drifts may occur, which can be eliminated through a high pass filter, singular spectrum analysis (SSA), or others. The high-pass filter is adopted herein for its simplicity. For field test use, these vehicle parameters can be calibrated by bump [39] or known-size hump test [49,50], to ensure accuracy of the desired level according to our previous studies in the field using the single-axle test vehicle [51].

As indicated by Equation (9), the *unknown input* to the test vehicle consists of two parts, i.e., *the contact displacement u* and *surface profile r*. To retrieve the surface profile r, a two-step procedure is proposed herein. The first is to estimate the unknown input u+r by the EKF-UI algorithm, and the second is to calculate the contact displacement u (which should be roughness profile free) and then deduct it from the unknown input u+r for retrieving the bridge surface profile r.

### 3.2. Step 1: Using the EKF-UI Algorithm to Estimate the Unknown Inputs

As indicated in Section 2, the state and measurement equations are required for estimating the unknown input. The entire procedure is outlined as follows.

#### 3.2.1. State-Space Equation for Test Vehicle Moving over Bridge

Let Z(t) denote the vehicle state vector:(10)$$Z\left(t\right)={\left[\begin{array}{cccc} {\dot{y}}_{1}\left(t\right) & {\dot{y}}_{2}\left(t\right) & y_{1}\left(t\right) & y_{2}\left(t\right) \end{array}\right]}^{T}$$

One can transform Equation (8) into the state space as
(11)$$\dot{Z}\left(t\right)=A_{c}Z\left(t\right)+B_{c}^{*}f^{*}\left(t\right)$$
where $A_{c}$ and $B_{c}^{*}$ can be calculated,
(12)$$A_{c}=\left[\begin{array}{cccc} 0 & 0 & -\frac{k_{1}\left(J_{v}+d_{1}^{2}m_{v}\right)}{m_{v}J_{v}} & -\frac{k_{2}\left(J_{v}-d_{1}d_{2}m_{v}\right)}{m_{v}J_{v}}\\ 0 & 0 & -\frac{k_{1}\left(J_{v}-d_{1}d_{2}m_{v}\right)}{m_{v}J_{v}} & -\frac{k_{2}\left(J_{v}+d_{2}^{2}m_{v}\right)}{m_{v}J_{v}}\\ 1 & 0 & 0 & 0\\ 0 & 1 & 0 & 0 \end{array}\right]\;B_{c}^{*}=\left[\begin{array}{cc} \frac{k_{1}\left(J_{v}+d_{1}^{2}m_{v}\right)}{m_{v}J_{v}} & \frac{k_{2}\left(J_{v}-d_{1}d_{2}m_{v}\right)}{m_{v}J_{v}}\\ \frac{k_{1}\left(J_{v}-d_{1}d_{2}m_{v}\right)}{m_{v}J_{v}} & \frac{k_{2}\left(J_{v}+d_{2}^{2}m_{v}\right)}{m_{v}J_{v}}\\ 0 & 0\\ 0 & 0 \end{array}\right]$$

Considering that the vehicle accelerations recorded are discrete in nature, one can discretize the state-space equation in Equation (11) as follows:(13)$$Z_{k+1}=\left(I+T_{s}A_{c}\right)Z_{k}+T_{s}B_{c}^{*}f_{k}^{*}=AZ_{k}+B^{*}f_{k}^{*}$$

Correspondingly, the discrete expressions of $Z_{k}$ and $f_{k}^{*}$ are
(14)$$Z_{k}={\left[\begin{array}{cccc} {\dot{y}}_{1}\left(k\right) & {\dot{y}}_{2}\left(k\right) & y_{1}\left(k\right) & y_{2}\left(k\right) \end{array}\right]}^{T}\;f_{k}^{*}=\left[\begin{array}{c} u_{1}\left(k\right)+r\left(k\right)\\ u_{2}\left(k\right)+r\left(k-\frac{d}{vT_{s}}\right) \end{array}\right]$$
where $T_{s}$ is the sampling interval and *k* the *k*th sampling point, k=1,2,⋯,n.

#### 3.2.2. Measurement Equation for Test Vehicle Moving over Bridge

Let $y_{k}$ denote the measurement vector for the test vehicle:(15)$$y_{k}={\left\{\begin{array}{cccc} {\ddot{y}}_{1}\left(k\right) & y_{1}\left(k\right) & {\ddot{y}}_{2}\left(k\right) & y_{2}\left(k\right) \end{array}\right\}}^{T}$$

The measurement equation for the vehicle can be derived from Equation (7) as
(16)$$y_{k}=C_{k}Z_{k}+D_{k}^{*}f_{k}^{*}+v_{k}$$
where $v_{k}$ is the noise vector, $C_{k}$ and $D_{k}^{*}$ are given as
(17)$$C_{k}\equiv \left[\begin{array}{cccc} 0 & 0 & -\frac{k_{1}\left(J_{v}+d_{1}^{2}m_{v}\right)}{m_{v}J_{v}} & -\frac{k_{2}\left(J_{v}-d_{1}d_{2}m_{v}\right)}{m_{v}J_{v}}\\ 0 & 0 & 1 & 0\\ 0 & 0 & -\frac{k_{1}\left(J_{v}-d_{1}d_{2}m_{v}\right)}{m_{v}J_{v}} & -\frac{k_{2}\left(J_{v}+d_{2}^{2}m_{v}\right)}{m_{v}J_{v}}\\ 0 & 0 & 0 & 1 \end{array}\right]\;D_{k}^{*}\equiv \left[\begin{array}{cc} \frac{k_{1}\left(J_{v}+d_{1}^{2}m_{v}\right)}{m_{v}J_{v}} & \frac{k_{2}\left(J_{v}-d_{1}d_{2}m_{v}\right)}{m_{v}J_{v}}\\ 0 & 0\\ \frac{k_{1}\left(J_{v}-d_{1}d_{2}m_{v}\right)}{m_{v}J_{v}} & \frac{k_{2}\left(J_{v}+d_{2}^{2}m_{v}\right)}{m_{v}J_{v}}\\ 0 & 0 \end{array}\right]$$

It should be noted that the vehicle responses in Equation (15) are those measured in the field, which may be polluted by noise, including ambient vibrations. To this end, one can employ the EKF-UI algorithm (Step 5) in Section 2 to obtain the improved state vector ${\hat{Z}}_{k+1|k+1}$. In other words, by using the state and measurement equations established for the vehicle in Equations (13) and (16), one can estimate from the recorded responses ${\ddot{y}}_{1}\left(k\right)$ and ${\ddot{y}}_{2}\left(k\right)$ the improved vehicle state vector ${\hat{Z}}_{k+1|k+1}$ and unknown input vector $f_{k}^{*}$ by the EKF-UI procedure described in Equations (A1)–(A10) and Figure 1.

### 3.3. Step 2: Calculation of Bridge Displacements at Contact Points

The state-space and measurement equations derived above allow us to estimate the unknown input $f_{k}^{*}$ in Equation (14), which includes both the surface profile r and contact displacement u of the bridge. In this regard, contact displacements $u_{1}\left(k\right)$ or $u_{2}\left(k\right)$ obtained should be roughness-free, such that it can be deducted from the unknown input $f_{k}^{*}$ to yield the bridge surface profile. Note that the bridge displacements used are those calculated from the improved (noise-reduced) vehicle state vector ${\hat{Z}}_{k}$ generated by the EKF-UI algorithm.

After some operations, one can obtain from Equation (11) the following equations:(18)$$\begin{array}{c} k_{1}\left\{\left[u_{1}\left(t\right)+{r\left(x\right)|}_{x=vt}\right]\right\}+k_{2}\left\{\left[u_{2}\left(t\right)+{r\left(x-d\right)|}_{x=vt}\right]\right\}\\ =\frac{m_{v}}{d}\left[d_{1}{\ddot{y}}_{2}\left(t\right)+d_{2}{\ddot{y}}_{1}\left(t\right)\right]+k_{1}y_{1}\left(t\right)+k_{2}y_{2}\left(t\right) \end{array}\;\begin{array}{c} d_{1}k_{1}\left[u_{1}\left(t\right)+{r\left(x\right)|}_{x=vt}\right]-d_{2}k_{2}\left[u_{2}\left(t\right)+{r\left(x-d\right)|}_{x=vt}\right]\\ =\frac{J_{v}}{d}\left[d_{1}{\ddot{y}}_{1}\left(t\right)-{\ddot{y}}_{2}\left(t\right)\right]+d_{1}k_{1}y_{1}\left(t\right)-d_{2}k_{2}y_{2}\left(t\right) \end{array}$$

Then, by performing the operations to Equation (18): “the *first* one”$\times d_{2}+$“the *second* one”, “the *first* one”$\times \left(-d_{1}\right)+$“the *second* one”, one can arrive at the following equations:(19)$$u_{1}\left(t\right)+{r\left(x\right)|}_{x=vt}=\frac{m_{v}d_{2}^{2}+J_{v}}{k_{1}d^{2}}{\ddot{y}}_{1}\left(t\right)+\frac{m_{v}d_{1}d_{2}-J_{v}}{k_{1}d^{2}}{\ddot{y}}_{2}\left(t\right)+y_{1}\left(t\right)$$
$$u_{2}\left(t\right)+{r\left(x-d\right)|}_{x=vt}=\frac{m_{v}d_{1}d_{2}-J_{v}}{k_{2}d^{2}}{\ddot{y}}_{1}\left(t\right)+\frac{m_{v}d_{1}^{2}+J_{v}}{k_{2}d^{2}}{\ddot{y}}_{2}\left(t\right)+y_{2}\left(t\right)$$

Further, by shifting the *second* one of Equation (19) by a time lag d/v or spatially by *d*, one obtains
(20)$$u_{2}\left(t+\frac{d}{v}\right)+{r\left(x\right)|}_{x=vt}=\frac{m_{v}d_{1}d_{2}-J_{v}}{k_{2}d^{2}}{\ddot{y}}_{1}\left(t+\frac{d}{v}\right)+\frac{m_{v}d_{1}^{2}+J_{v}}{k_{2}d^{2}}{\ddot{y}}_{2}\left(t+\frac{d}{v}\right)+y_{2}\left(t+\frac{d}{v}\right)$$

The first and second in Equation (19) represent exactly the *contact responses* of the front and rear axles, respectively, over the same location x of the bridge, implying that the same profile ${r\left(x\right)|}_{x=vt}$ was experienced by the two axles at different moments, i.e., t for the front axle, and t+d/v for the rear axle. 

Subtracting Equation (20) from the *first* one of Equation (19) yields the *residual response* ∆u(t) of the two contact points at the same location x of the bridge:(21)$$\begin{array}{c} ∆u\left(t\right)=u_{1}\left(t\right)-u_{2}\left(t+\frac{d}{v}\right)=\frac{m_{v}d_{2}^{2}+J_{v}}{k_{1}d^{2}}{\ddot{y}}_{1}\left(t\right)-\frac{m_{v}d_{1}d_{2}-J_{v}}{k_{2}d^{2}}{\ddot{y}}_{1}\left(t+\frac{d}{v}\right)\\ \;+\frac{m_{v}d_{1}d_{2}-J_{v}}{k_{1}d^{2}}{\ddot{y}}_{2}\left(t\right)-\frac{m_{v}d_{1}^{2}+J_{v}}{k_{2}d^{2}}{\ddot{y}}_{2}\left(t+\frac{d}{v}\right)+y_{1}\left(t\right)-y_{2}\left(t+\frac{d}{v}\right) \end{array}$$

Evidently, the unknown profile ${r\left(x\right)|}_{x=vt}$ has been eliminated from Equation (21) by *subtraction*, and all the remaining terms are those known of the test vehicle. In this sense, the contact residual response ∆u(t) is said to be *roughness free*. It can be easily obtained in field tests or by numerical simulation. By noting that the recorded vehicle accelerations are *discrete* in nature, Equation (21) can also be recast in discrete form:(22)$$\begin{array}{l} ∆u\left(k\right)=u_{1}\left(k\right)-u_{2}\left(k+\frac{d}{vT_{s}}\right)=\frac{m_{v}d_{2}^{2}+J_{v}}{k_{1}d^{2}}{\ddot{y}}_{1}\left(k\right)-\frac{m_{v}d_{1}d_{2}-J_{v}}{k_{2}d^{2}}{\ddot{y}}_{1}\left(k+\frac{d}{vT_{s}}\right)\\ \;+\frac{m_{v}d_{1}d_{2}-J_{v}}{k_{1}d^{2}}{\ddot{y}}_{2}\left(k\right)-\frac{m_{v}d_{1}^{2}+J_{v}}{k_{2}d^{2}}{\ddot{y}}_{2}\left(k+\frac{d}{vT_{s}}\right)+y_{1}\left(k\right)-y_{2}\left(k+\frac{d}{vT_{s}}\right) \end{array}$$

With this, the *cumulated contact residuals*, $\sum_{i=1}^{k}∆u\left(k\right)$, that is roughness-free, can be calculated as well. For the present purposes, one assumes a priori that there exists a correlation between $\sum_{i=1}^{k}∆u\left(k\right)\;$. and *u*(*k*), i.e.,
(23)$$\varphi_{k}=\frac{\sum_{i=1}^{k}∆u\left(k\right)}{u_{1}\left(k\right)}\;\lambda_{k}=\frac{\sum_{i=1}^{k}∆u\left(k\right)}{u_{2}\left(k\right)}$$

Though the numerator $\sum_{i=1}^{k}∆u\left(k\right)$ is known, the contact displacement u(k) in the denominator of Equation (23) is unknown due to the involvement of the flexural rigidity EI and frequency $\omega_{bn}$ of the bridge, which renders the coefficients $\varphi_{k}$ and $\lambda_{k}$ not readily available. Obviously, to retrieve the contact displacements u(k), the correlation between $\sum_{i=1}^{k}∆u\left(k\right)$ and u(k), i.e., the coefficients $\varphi_{k}$ and $\lambda_{k}$, should be determined first, as will first be explained in the following.

For a two-axle vehicle traveling over a simple beam, the vehicle–bridge contact (point) displacements are [52]:(24)$$\begin{array}{c} u_{i}\left(t\right)=\underset{N}{\sum }\sum_{j=1}^{2}\frac{A_{n}p_{j}}{1-S_{n}^{2}}\sin\left[\frac{n\pi v\left(t-t_{i}\right)}{L}\right]\left\{\sin\left[\frac{n\pi v\left(t-t_{j}\right)}{L}\right]-S_{n}\sin\omega_{bn}\left(t-t_{j}\right)\right\}\\ \times H\left(t-t_{j}\right)\;\;\;\;i=1,2 \end{array}$$
where H(·) is the unit step function; *N* the number of modes considered; $A_{n}$ the nth modal (equivalent) static deflection of the bridge; $S_{n}$ the nth frequency ratio of the driving frequency nπv/L to the bridge frequency $\omega_{bn}$; $p_{j}$ the jth axle load of the vehicle; $t_{j}$ the time for jth axle to enter the bridge; namely,
(25)$$A_{n}=-\frac{2L^{3}}{EIn^{4}\pi^{4}},S_{n}=\frac{n\pi v}{L\omega_{bn}},\omega_{bn}=\frac{n^{2}\pi^{2}}{L^{2}}\sqrt{\frac{EI}{m}}\;p_{j}=\frac{d-d_{j}}{d}m_{v}g\;\mathrm{for}\;j=1,2\;t_{j}=\frac{\left(j-1\right)d}{v}\;\mathrm{for}\;j=1,2$$

When using the test vehicle to measure the bridge surface roughness, the test speed should be kept reasonably low to avoid the vehicle’s separation from the bridge on the one hand, while ensuring a sufficient amount of data being collected on the other hand. In this situation, the driving frequency nπv/L used is generally much less than the bridge frequency $\omega_{bn}$, i.e., $S_{n}\to 0$. Consequently, the contact displacements in Equation (24) are reduced to
(26)$$u_{i}\left(t\right)\approx u_{i}^{s}\left(t\right)=\underset{N}{\sum }\sum_{j=1}^{2}A_{n}p_{j}\sin\left[\frac{n\pi v\left(t-t_{i}\right)}{L}\right]\times \sin\left[\frac{n\pi v\left(t-t_{j}\right)}{L}\right]\times H\left(t-t_{j}\right)\;i=1,2$$
which can be expanded and discretized to yield the *contact displacements* $u_{1}$ and $u_{2}$ of the two axles, along with the shifted response for rear axle, as follows:(27)$$\begin{array}{cc} u_{1}\left(k\right)\approx u_{1}^{s}\left(k\right) & =\underset{N}{\sum }A_{n}\sin\left(\frac{n\pi vkT_{s}}{L}\right)\;\;\;\;\;\;\;\\ & \times \left[p_{1}\sin\left(\frac{n\pi vkT_{s}}{L}\right)+p_{2}\sin\left(\frac{n\pi v\left(kT_{s}-d/v\right)}{L}\right)\times H\left(kT_{s}-d/v\right)\right]\\ u_{2}\left(k\right)\approx u_{2}^{s}\left(k\right) & =\underset{N}{\sum }A_{n}\sin\left(\frac{n\pi v\left(kT_{s}-d/v\right)}{L}\right)\times H\left(kT_{s}-d/v\right)\\ & \times \left[p_{1}\sin\left(\frac{n\pi vkT_{s}}{L}\right)+p_{2}\sin\left(\frac{n\pi v\left(kT_{s}-d/v\right)}{L}\right)\times H\left(kT_{s}-d/v\right)\right]\\ u_{2}\left(k+\frac{d}{vT_{s}}\right) & \approx u_{2}^{s}\left(k+\frac{d}{vT_{s}}\right)=\underset{N}{\sum }A_{n}\sin\left(\frac{n\pi vkT_{s}}{L}\right)\\ & \times \left[p_{2}\sin\left(\frac{n\pi vkT_{s}}{L}\right)+p_{1}\sin\left(\frac{n\pi v\left(kT_{s}+d/v\right)}{L}\right)\times H\left(\frac{L-d}{v}-kT_{s}\right)\right] \end{array}$$

By substituting Equation (27) into Equation (23), one obtains approximate expressions for the coefficients $\varphi_{k}$ and $\lambda_{k}$, i.e., $\varphi_{k}^{s}$ and $\lambda_{k}^{s}$, as follows:(28)$$\varphi_{k}\approx \varphi_{k}^{s}=\frac{\sum_{i=1}^{k}\left[u_{1}^{s}\left(k\right)-u_{2}^{s}\left(k+d/vT_{s}\right)\right]}{u_{1}^{s}\left(k\right)}\;\lambda_{k}\approx \lambda_{k}^{s}=\frac{\sum_{i=1}^{k}\left[u_{1}^{s}\left(k\right)-u_{2}^{s}\left(k+d/vT_{s}\right)\right]}{u_{2}^{s}\left(k\right)}$$

It is interesting to note that the flexural rigidity *EI* is the only property of the bridge involved in Equation (28) for calculating the contact displacements, and that it will be canceled out since it appears both in the numerator and denominator. Consequently, *the coefficients*
$\varphi_{k}$
*and*
$\lambda_{k}$
*depend only on the vehicle’s parameters and location on the bridge*, all of which are known during the test. In other words, the coefficients $\varphi_{k}$ and $\lambda_{k}$ can be readily made available for each test vehicle. By the way, Equation (28) is also applicable to the case with a slight local reduction in flexural rigidity via the involvement of bridge deflections [53]. It should be recalled that the quantity $\sum_{i=1}^{k}∆u\left(k\right)\;$was already made available via the use of Equation (22). Inasmuch as one can calculate the vehicle–bridge contact displacements by substituting Equation (28) into Equation (23) to yield $u_{1}\left(k\right)=\frac{\sum_{i=1}^{k}∆u\left(k\right)}{\varphi_{k}}$ and $u_{2}\left(k\right)=\frac{\sum_{i=1}^{k}∆u\left(k\right)}{\lambda_{k}}$, which are also roughness free. The reliability of the above procedure for calculating the contact displacements will be verified by the FEM in Section 4. Finally, the surface profile r(k) of the bridge can be recovered by deducting u(k) from the estimated input $f_{k}^{*}$.

### 3.4. Flowchart for Retrieval of Bridge Profile

A summary of the proposed technique is given in the flowchart of Figure 3. First, through the EKF-UI algorithm, both the unknown input vector $f_{k}^{*}$ and the improved (noise-reduced) vehicle responses ${\hat{Z}}_{k}$ can be estimated, and the latter are used to calculate the cumulative contact residual $\sum_{i=1}^{k}∆u\left(k\right)$. Second, the contact (bridge) displacements u(k) are estimated by using $\sum_{i=1}^{k}∆u\left(k\right)$ (which are roughness free) and the coefficients $\varphi_{k}^{s}$ and $\lambda_{k}^{s}$, as given in Equation (28). Finally, by deducting the contact displacements u(k) from the estimated $f_{k}^{*}$, one obtains the bridge surface profile r(k).

As for multi-axle vehicles, the unknown inputs to the test vehicle should be extended to the motions of all contact points. Accordingly, the state vector and equation and the measurement vector and equation should also be extended to all the axles. The contact displacements can be similarly calculated by the procedure presented in Section 3.3.

## 4. Numerical Validation of the Proposed Procedure

Numerical validation consists of two phases that are of completely different algorithms and were executed by two independent groups within our research team. (1) Forward phase: The random original (or input) surface profile generated (see Section 4.1) will be included in the VBI calculation program (See Section 4.2) to simulate the vehicle response for the test vehicle drive over a rough bridge. Then, the Gaussian white noise will be superimposed on the vehicle responses to simulate the (measured) noise-contaminated vehicle response (see Section 4.3). (2) Backward phase: The proposed two-step technique will be employed on the noise-contaminated vehicle response created in the forward phase to estimate the improved (noise-reduced) vehicle states and surface profile, with which the efficacy will be verified (see Section 4.4).

### 4.1. Generation of Bridge Surface Profile

In practical measurement, the bridge surface roughness is unknown and can be estimated using the procedure presented herein. However, in this numerical simulation, for the purpose of verification, the surface roughness will be assumed to be known, and the roughness profile estimated by the proposed procedure will be compared with the known one to assess the level of accuracy.

In the ISO 8608 standard, the road surface profile *r*(*x*) is expressed as the superposition of a series of trigonometric functions via the power spectrum density (PSD) function as [54]:(29)$$r\left(x\right)=\sum_{N}\sqrt{2G\left(n_{i}\right)∆n}\cos\left(2\pi n_{i}x+\theta_{i}\right)$$
where *x* is the distance along the bridge, ∆n the average frequency increment, $\theta_{i}$ the random phase angle uniformly distributed in [0, 2π], and $G\left(n_{i}\right)$ is the PSD function,
(30)$$G\left(n_{i}\right)=G\left(n_{0}\right){\left(\frac{n_{i}}{n_{0}}\right)}^{-2}$$

Here $G\left(n_{0}\right)$ is assigned values for different classes of roughness, A, B, C, D and E (from the best to the poorest), as listed in Table 1.

### 4.2. Finite Element Method for Generating VBI Responses

For the two-axle vehicle passing a bridge, it can be modeled by the VBI element shown in Figure 4, where the axle interval *d* is assumed to be greater than the element length *l_e_*, and the front and rear axles (wheels) are in contact with elements *i* and *j* at points P_1_ and P_2_, respectively. Considering the bridge surface roughness, the equation of motion for the VBI element can be expressed as:(31)$$\left[\begin{array}{cc} M_{b} & 0\\ 0 & M_{v} \end{array}\right]\left\{\begin{array}{c} {\ddot{x}}_{b}\\ {\ddot{x}}_{v} \end{array}\right\}+\left[\begin{array}{cc} C_{b} & 0\\ 0 & 0 \end{array}\right]\left\{\begin{array}{c} {\dot{x}}_{b}\\ {\dot{x}}_{v} \end{array}\right\}+\left[\begin{array}{cc} K_{b} & 0\\ 0 & K_{v} \end{array}\right]\left\{\begin{array}{c} x_{b}\\ x_{v} \end{array}\right\}=\left\{\begin{array}{c} F_{b}\\ F_{v} \end{array}\right\}$$
where **x** denotes the response vectors, **M**, **C** and **K**, respectively, the mass, damping and stiffness matrices, F the force vectors, and the subscript *b* for bridge and *v* for vehicle. The matrices in Equation (31) are given in Appendix B.

By assembling the above VBI element and the conventional elements for the parts of the beam that are not in direct contact with the vehicle, the equation of motion for the entire system can be established. Then, by applying the Newmark-*β* method (with *α* = 0.25 and *β* = 0.5), the dynamic responses of the VBI system can be solved.

### 4.3. Simulation of (Measured) Noise-Contaminated Vehicle Responses

Environmental noise may pollute the data collected by the moving test vehicle and therefore reduce the measurement accuracy of the vehicle scanning method for bridges. To simulate such an effect, Gaussian white noise will be superimposed on the calculated vehicle responses, i.e.,
(32)$$y_{k}^{p}=y_{k}+E_{p}N_{s}\sigma_{y_{k}}$$
where $y_{k}$ and $y_{k}^{p}$ denote the original and polluted responses of the test vehicle, respectively, $N_{s}$ the standard normal distribution, $\sigma_{y_{k}}$ the standard deviation of $y_{k}$, and $E_{p}$ the noise level.

### 4.4. Validation of the Proposed Procedure

For the sake of comparison, the properties of the bridge and vehicle used by Yang et al. [52] were adopted in the present study, as listed in Table 2. As mentioned previously, the proposed procedure does not depend on the flexural stiffness *EI* of the bridge. For simplicity, the *EI* value was taken to be uniform for the beam. With reference to Figure 4, the properties of the test vehicle are: mass $m_{v}=2500\;\mathrm{kg}$, moment of inertia $J_{v}=2300\;\mathrm{kg}\cdot m^{2}$, axle distances to center of gravity $d_{1}=1.7\;m$ and $d_{2}=1.3\;m$, and axle suspension stiffness $k_{1}=230\;\mathrm{kN}\cdot m^{-1},k_{2}=180\;\mathrm{kN}\cdot m^{-1}$. Vehicle speed is set at *v* = 2 m/s and time step is 0.001s. Meanwhile, the following initial values are adopted for the EKF-UI: ${\hat{Z}}_{0|0}={\left[0\;0\;0\;0\right]}^{T},$ ${\hat{f}}_{0|0}^{*}={\left[0\;0\;0\;0\right]}^{T}$, $P_{Z,0|0}=\mathrm{diag}\left[1\;1\;10^{6}\;10^{6}\;\right]$, $Q=10^{-8}I_{4}$, and $R=10^{-3}I_{4}$, where $I_{4}$ denotes the (4 × 4) identity matrix. In addition, a noise of $E_{p}=2\%\;$is added to the calculated vehicle response through Equation (32) to simulate the noise-contaminated effect.

As revealed in the above formulation, to guarantee the accuracy of the retrieved bridge surface profile, two issues are considered essential. One is the improved (noise-reduced) vehicle states that may affect the accuracy of the cumulative contact residual $\sum_{i=1}^{k}∆u\left(k\right)$. The other is the coefficients $\varphi_{k}^{s}$ and $\lambda_{k}^{s}$ used to determine the bridge displacements u(k). For this reason, the estimation for the improved vehicle states and the coefficients $\varphi_{k}^{s}$ and $\lambda_{k}^{s}$ should be validated first.

#### 4.4.1. Validation of the Estimation for the Improved Vehicle States

As indicated by Equation (A9), the improved (noise-reduced) vehicle state can be obtained recursively from the contaminated measured response using the EKF-UI algorithm. The purpose herein is to validate the result retrieved from the vehicle state, including the velocity and displacement responses of the two axles, as defined in Equation (15). To validate the applicability of the proposed procedure to various roughness qualities, both roughness Classes A and C are selected to represent good and poor surface conditions, respectively. For the two-axle vehicle moving over the surface roughness of Classes A and C, the axle responses retrieved from the noise-contaminated vehicle response by the EKF-UI algorithm were compared with the original responses (with no noise) in Figure 5, Figure 6, Figure 7 and Figure 8, where parts (a) and (b) denote the responses of the two axles. As revealed by the figures, all the retrieved responses match well with the original ones, confirming that *the EKF-UI algorithm is effective for removing the measurement noise from the measured vehicle responses* through the recursive procedure.

#### 4.4.2. Validation of the Coefficients $\varphi_{k}^{s}$ and $\lambda_{k}^{s}$

In the preceding section, the coefficients $\varphi_{k}^{s}$ and $\lambda_{k}^{s}$ have been theoretically derived for the calculation of the contact displacements and then for retrieval of the bridge surface profile. In this section, the efficacy of such a procedure will be verified numerically.

Based on Equations (23) and (28), one can retrieve the contact responses as follows:(33)$$u_{1}\left(k\right)\approx \frac{\sum_{i=1}^{k}∆u\left(k\right)}{\varphi_{k}^{s}}\;u_{2}\left(k\right)\approx \frac{\sum_{i=1}^{k}∆u\left(k\right)}{\lambda_{k}^{s}}$$

Herein, it is emphasized that cumulative contact residual $\sum_{i=1}^{k}∆u\left(k\right)$ is computed from the vehicle states $y_{1}$ and $y_{2}$ based on Equation (22), and the latter are generated as the improved state vector by the EKF-UI, as shown in Equation (A9). The contact responses $u_{1}^{r}$ and $u_{2}^{r}$ retrieved from Equation (33) for the front and rear axles are shown in Figure 9a,b, respectively, along with those by the FEM using the given properties of the VBI system. As can be seen, the retrieved contact responses $u_{1}^{r}$ and $u_{2}^{r}$ agree well with the original ones $u_{1}$ and $u_{2}$, respectively. This example demonstrates the efficacy of the coefficient $\varphi_{k}^{s}$ and $\lambda_{k}^{s}$ in combination with the use of the EKF-UI algorithm.

#### 4.4.3. Validation of the Estimation for the Bridge Surface Profile

In this section, the bridge surface profile will be retrieved by use of the proposed two-step technique and the result will be compared with the original (assumed) input. The surface profiles of Classes A and C roughness for the simple beam retrieved by the EKF-UI algorithm have been plotted in Figure 10 and Figure 11, respectively, along with the original (assumed) ones (generated by the PSD). In the figures, parts (a) and (b) denote the results in the spatial and frequency domains, respectively. It is confirmed that regardless of the varying class of roughness, the retrieved profile is in good agreement with the original profile in both the spatial and frequency domains. This indicates *the reliability of the proposed technique for retrieving the bridge surface profile.* It should be added that for all the cases presented in this study, the computation time for the proposed technique using typical notebooks is in seconds, meaning that the computational expense is not a problem of concern.

## 5. Parametric Study

In the field test, factors such as the moving speed of the test vehicle, vehicle mass, environmental noise, bridge damping, etc., may affect the VBI responses and further the retrieval of the bridge surface profile. To this end, a parametric study will be conducted to evaluate the capability of the proposed procedure against these factors. For simplicity, only Class A roughness is considered. The error indicator RMSE (Root Mean Square Error) [42] is adopted for estimating the accuracy of the solution:(34)$$\mathrm{RMSE}=\frac{1}{x_{o,\max}}\sqrt{\frac{1}{N}\sum_{i=1}^{N}{\left(x_{r,\;i}-x_{o,\;i}\right)}^{2}}$$
where $x_{o,\;i}$ and $x_{r,\;i}$ respectively denote the ith original (generated by the PSD) and retrieved values of the time series *x*; $x_{o,\max}$ the maxmum absolute value of the original sreies; and *N* the total number of data points. 

### 5.1. Effect of Vehicle Speed

In Section 4, it was verified that the proposed method is effective in estimating bridge roughness at a low speed of 2 m/s. In this section, the effect of higher speeds will be studied by including a medium speed of 8 m/s and a high speed of 16 m/s. The deviations of the retrieved response from the input bridge surface profile are plotted in Figure 12, which are all quite small. Meanwhile, the error indicator RMSEs calculated for the three speeds of 2, 8 and 16 m/s are 3.66, 4.84 and 4.91%, respectively.

From these figures, even at higher speeds, the retrieved results of the bridge profile are in good agreement with the original ones. However, the error indicator RMSE increases from 3.66 to 4.91% for a vehicle speed increasing from 2 to 16 m/s, implying a decrease in the estimation accuracy of the bridge profile for increasing speed. Nevertheless, *for vehicle speeds lower than 16 m/s, the errors of the retrieved profile are lower than 5%*, which is acceptable for practical use. Therefore, it is suggested that a vehicle speed lower than 16 m/s be used.

### 5.2. Effect of Vehicle–Bridge Mass Ratio

In Section 4, it was shown that the surface profile can be accurately retrieved for the vehicle mass of 2500 kg (mass ratio $m_{v}/mL=$ 4.17%). In reality, for the portability and mobility of test vehicles, they are designed to be even lighter. To reflect this concern, three more vehicle masses, i.e., 1500, 2000, 3000 kg (with mass ratios 2.50, 3.33, 5.00%), are considered in this section to study their effect on the proposed technique.

Four mass ratios of 2.50, 3.33, 4.17, and 5.00%, the deviations of the retrieved response from the input bridge surface profile are plotted in Figure 13, and the calculated error indicator RMSEs were 4.09, 4.52, 3.66 and 4.43%, respectively. No clear relationship exists between the retrieved profiles and mass ratios. The main reason is that a higher vehicle mass can induce a higher bouncing impact on the bridge [1], and therefore a larger error for the coefficient $\varphi_{k}^{s}$ and $\lambda_{k}^{s}$, and further the profile. However, heavier vehicles are not so sensitive to roughness, unlike lighter vehicles, the latter are likely to be set to violent vibrations, and accordingly heavier vehicles induce a lower error for the improved vehicle states and further for the roughness. Fortunately, all RMSEs of the retrieved profiles are below 5%, indicating that *the vehicle–bridge mass ratios have no obvious effect on the proposed technique for retrieving the bridge surface profile.*

### 5.3. Effects of Environmental Noise

To investigate the effect of environment noise on the proposed procedure, four noise levels, i.e., 0.00, 2.00, 5.00, and 10.00%, are considered herein. The deviations of the retrieved response from the input bridge surface profile are plotted in Figure 14, and the error indicator RMSEs for the four levels of noise are 2.88, 3.66, 7.24 and 14.14%, respectively. From these results, one can observe that the estimation accuracy of bridge profiles decreases with increasing noise levels. For a large noise level of *Ep* = 10%, the RMSE will reach 14%, indicating that the profile retrieved is sensitive to noise. It is suggested that *the proposed technique be conducted in an environment of low noise* to retrieve the bridge profile.

### 5.4. Effect of Bridge Damping

For the VBI system, damping may affect the transmissibility of vibrations and therefore the retrieval of the bridge profile. Vehicle’s damping is required to be as low as possible and in many cases adjustable, but bridge’s damping in fact can hardly be adjusted. As such, it is necessary to study the effect of the bridge damping on the proposed method. Three damping ratios of 1.00, 3.00 and 5.00% of the Rayleigh type are considered for the bridge. The deviations of the retrieved response from the input bridge surface profiles are plotted in Figure 15, and the corresponding error indicator RMSEs found are 4.33, 3.66 and 3.43%, respectively.

As can be seen, as the bridge damping increases, the error indicator RMSEs of the bridge surface profile gradually decrease, and all are less than 5%. This is due to the fact that the larger the bridge damping, the lower the vehicle-induced dynamic impact on the bridge, and therefore the higher the accuracy for the coefficient $\varphi_{k}^{s}$ and $\lambda_{k}^{s}$, and further the roughness. In other words, *the bridge surface profile can be well retrieved even in the presence of bridge damping*, and the larger the bridge damping, the better the accuracy of estimation.

## 6. Concluding Remarks

In this study, a two-step technique is proposed for retrieving the bridge surface profile in a noisy environment from the responses recorded of a two-axle test vehicle moving over the bridge. First, the EKF-UI algorithm is employed to estimate the improved (noise-reduced) vehicle states and the unknown inputs consisting of surface profile and contact displacement. Second, the contact displacements are calculated from the cumulated contact residuals (that is roughness-free) using the improved vehicle states and then deducted from the unknown input for retrieving the surface profile. Based on the theory and numerical studies presented in this paper, together with the properties adopted for the vehicle–bridge system, the following conclusions are drawn:

(1)The estimated vehicle states and bridge surface profiles by the two-step technique agree well with the original ones, which verifies the feasibility of the proposed procedure.(2)The coefficients $\varphi_{k}^{s}$ and $\lambda_{k}^{s}$ used to define the correlation between the cumulated contact residuals $\sum_{i=1}^{k}∆u\left(k\right)$ and the contact displacement u(k) of two axles are reliable, which can be accurately estimated without prior knowledge of the bridge dynamic properties.(3)The estimation accuracy of the bridge profile decreases with increasing vehicle speed. Nevertheless, for vehicle speeds lower than 16 m/s, the errors of the profile are lower than 5%, which is acceptable for practical use.(4)For the vehicle–bridge mass ratios considered, the estimated error RMSEs of the retrieved profile are all below 5%, indicating that the vehicle–bridge mass ratios have no obvious effect on the proposed technique for surface profile retrieval.(5)The profile is sensitive to environmental noise, as the RMSE will reach 14% for the noise level of *Ep* = 10%. It is thus suggested that the proposed technique be conducted in an environment of low noise to retrieve the bridge profile.(6)The bridge profile can be well retrieved in the presence of bridge damping. The larger the bridge damping, the better the accuracy of the profile estimation.

This paper has numerically validated the effectiveness of the proposed procedure in retrieving the surface roughness of bridges. Further work will be conducted on the application of the technique to the field, considering bridges of different types.

## Figures and Tables

**Figure 1 sensors-22-03410-f001:**
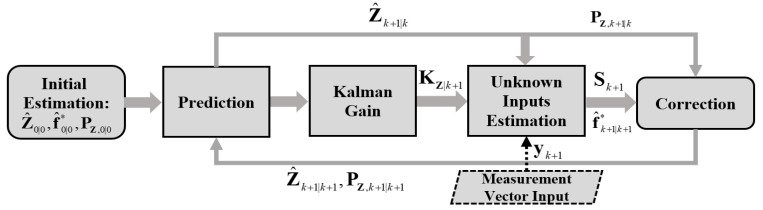
Flowchart of the EKF-UI algorithm.

**Figure 2 sensors-22-03410-f002:**
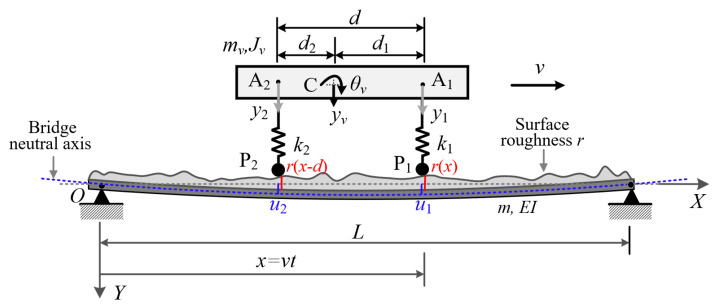
Vehicle–bridge interaction model with roughness.

**Figure 3 sensors-22-03410-f003:**
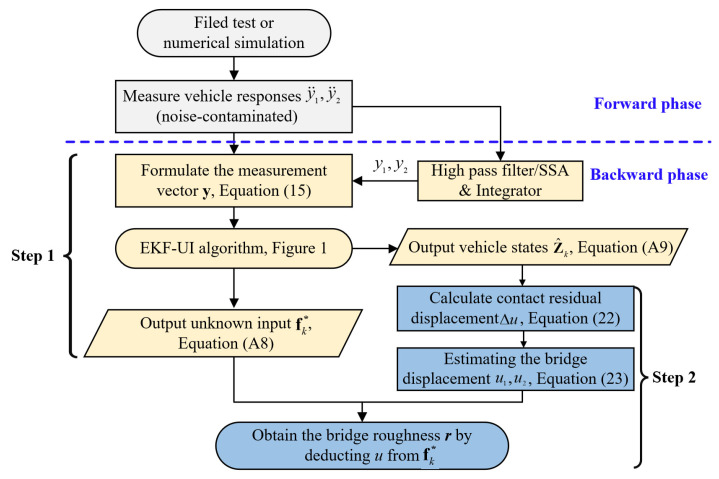
Flowchart of the proposed technique.

**Figure 4 sensors-22-03410-f004:**
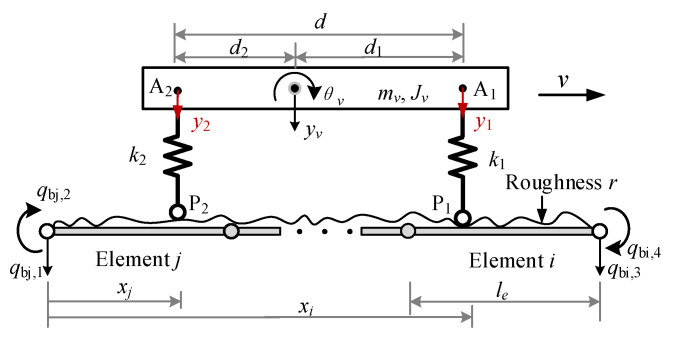
Two-axle vehicle–bridge interaction element.

**Figure 5 sensors-22-03410-f005:**
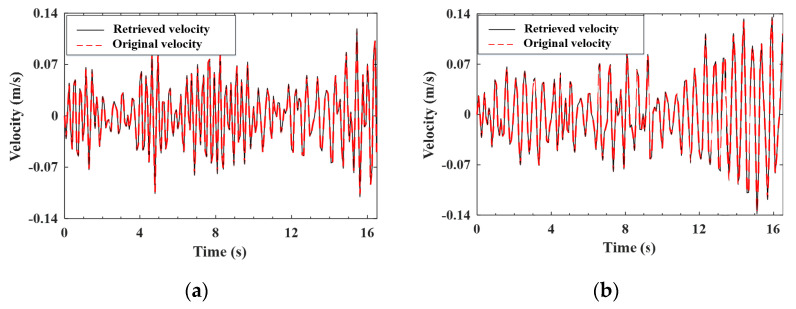
Retrieved and original velocity responses of the test vehicle (Class A roughness). (**a**) front axle; (**b**) rear axle.

**Figure 6 sensors-22-03410-f006:**
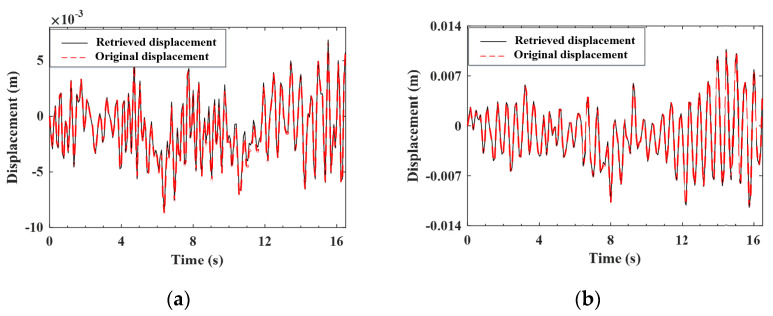
Retrieved and original displacement responses of the test vehicle (Class A roughness). (**a**) front axle; (**b**) rear axle.

**Figure 7 sensors-22-03410-f007:**
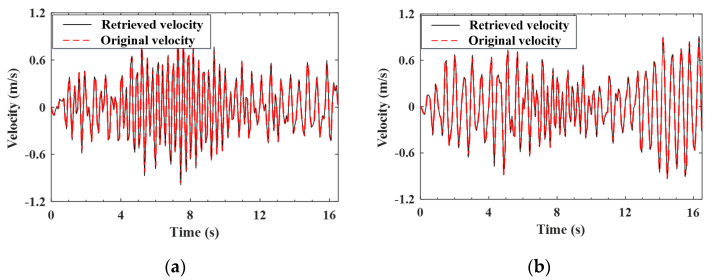
Retrieved and original velocity responses of the test vehicle (Class C roughness). (**a**) front axle; (**b**) rear axle.

**Figure 8 sensors-22-03410-f008:**
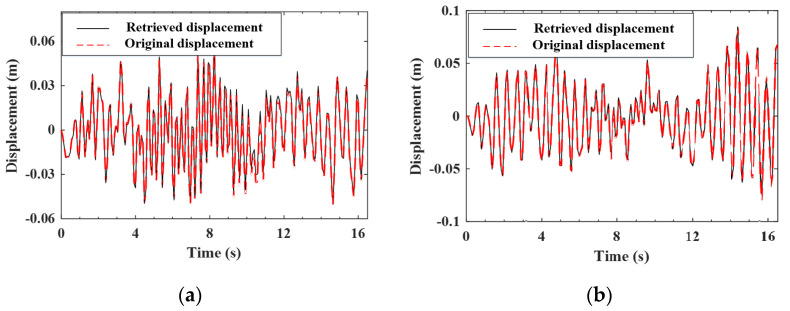
Retrieved and original displacement responses of the test vehicle (Class C roughness). (**a**) front axle; (**b**) rear axle.

**Figure 9 sensors-22-03410-f009:**
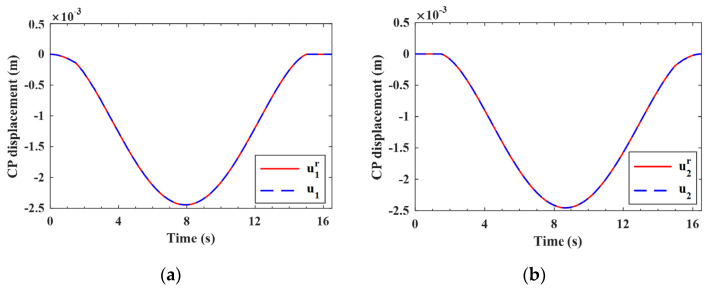
Contact displacements for the two-axle vehicle moving over the bridge. (**a**) front wheel; (**b**) rear wheel.

**Figure 10 sensors-22-03410-f010:**
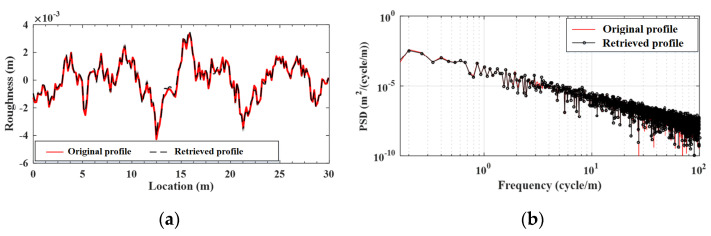
Retrieved and original surface profile (Class A roughness). (**a**) spatial domain; (**b**) frequency domain.

**Figure 11 sensors-22-03410-f011:**
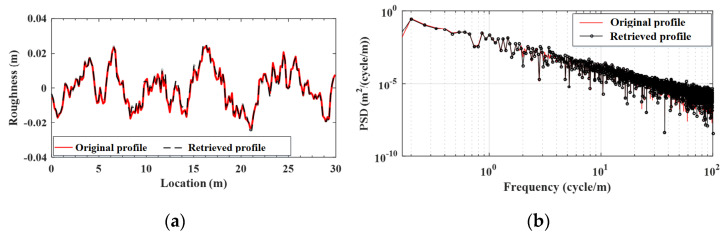
Retrieved and original surface profile (Class C roughness). (**a**) spatial domain; (**b**) frequency domain.

**Figure 12 sensors-22-03410-f012:**
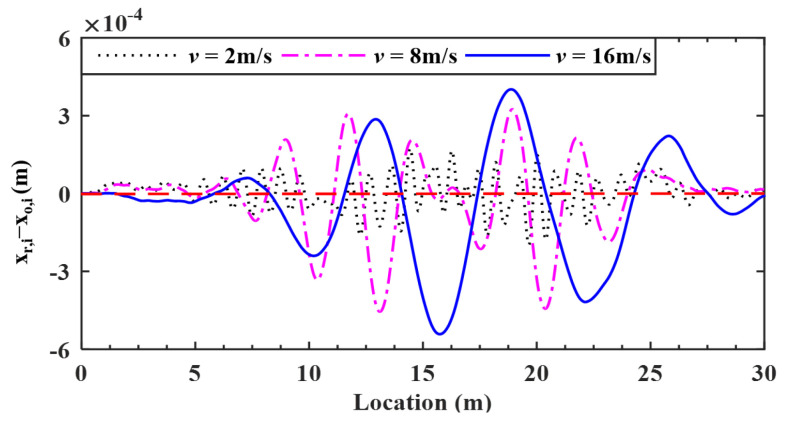
Deviations of surface profiles identified for different vehicle speeds.

**Figure 13 sensors-22-03410-f013:**
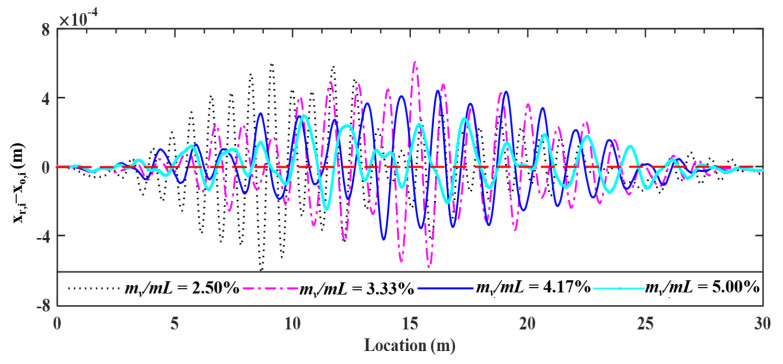
Deviations of surface profile identified for different vehicle masses.

**Figure 14 sensors-22-03410-f014:**
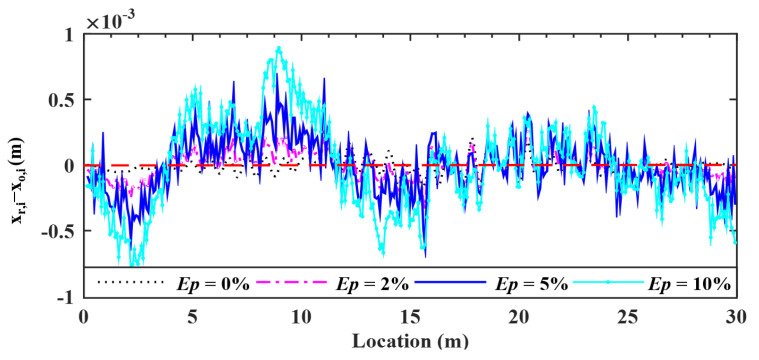
Deviations of surface profiles identified for different noise levels.

**Figure 15 sensors-22-03410-f015:**
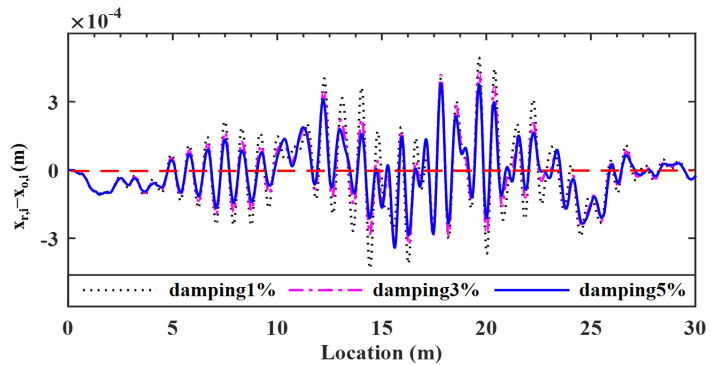
Deviations of surface profiles identified for different bridge damping ratios.

**Table 1 sensors-22-03410-t001:** Coefficient values $G\left(n_{0}\right)$ for different classes of roughness.

Roughness Class	Description	$G_{d}\left(n_{0}\right)\;(10^{-6}\;m^{3})$		
		Lower Limit	Geometric Mean	Upper Limit
A	Very good	-	16	32
B	Good	32	64	128
C	Average	128	256	512
D	Poor	512	1024	2048
E	Very poor	2048	4096	8192

**Table 2 sensors-22-03410-t002:** Physical properties of bridge.

Bridge Properties			
Young’s modulus	*E*	GPa	27.5
Moment of inertia	*I*	m^4^	0.2
Mass per unit length	*m*	kg·m^−1^	2000
Span length	*L*	m	30
Beam element length	*l_e_*	m	1.0
First modal damping ratio	*ξ* _1_	%	3.0
Second modal damping ratio	*ξ* _2_	%	3.0

## Data Availability

Data is contained within the article.

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
