# Peer review of "Scanning of Bridge Surface Roughness from Two-Axle Vehicle Response by EKF-UI and Contact Residual: Theoretical Study"

_sensors, 2022, doi:10.3390/s22093410_

Round 1

Reviewer 1 Report

Page 2

Define LIDAR and 3D in full form

Define ADAMS in full form

General comment

KF is an old method. Now researchers are using EKF, UKF, AKF, etc… which are based on more accuracy. I have doubts that KF is not an optimal solution. In this paper authors implemented only KF which is not an optimal solution. Authors should compare EKF, UKF, AKF etc… with KF then conclusion will be justify.

Reviewer 2 Report

Please find my comments in the attached document. 

Round 2

Reviewer 1 Report

Paper is improved.

Reviewer 2 Report

I am happy with the review responses and I would like to accept the paper at this stage. 

This manuscript is a resubmission of an earlier submission. The following is a list of the peer review reports and author responses from that submission.